# Potential Therapeutic Targets to Modulate the Endocannabinoid System in Alzheimer’s Disease

**DOI:** 10.3390/ijms25074050

**Published:** 2024-04-05

**Authors:** Hina Kanwal, Moris Sangineto, Martina Ciarnelli, Pasqualina Castaldo, Rosanna Villani, Antonino Davide Romano, Gaetano Serviddio, Tommaso Cassano

**Affiliations:** 1Department of Medical and Surgical Sciences, University of Foggia, 71122 Foggia, Italy; moris.sangineto@unifg.it (M.S.); martina.ciarnelli@unifg.it (M.C.); rosanna.villani@unifg.it (R.V.); antonino.romano@unifg.it (A.D.R.); gaetano.serviddio@unifg.it (G.S.); tommaso.cassano@unifg.it (T.C.); 2Department of Biomedical Sciences and Public Health, School of Medicine, University “Politecnica delle Marche”, 60126 Ancona, Italy; p.castaldo@univpm.it

**Keywords:** neurodegenerative diseases, Alzheimer’s disease, endocannabinoid system, fatty acid amide hydrolase (FAAH), monoacylglycerol (MAGL), cannabinoid receptors (CB1R; CB2R)

## Abstract

Alzheimer’s disease (AD), the most common neurodegenerative disease (NDD), is characterized by chronic neuronal cell death through progressive loss of cognitive function. Amyloid beta (Aβ) deposition, neuroinflammation, oxidative stress, and hyperphosphorylated tau proteins are considered the hallmarks of AD pathology. Different therapeutic approaches approved by the Food and Drug Administration can only target a single altered pathway instead of various mechanisms that are involved in AD pathology, resulting in limited symptomatic relief and almost no effect in slowing down the disease progression. Growing evidence on modulating the components of the endocannabinoid system (ECS) proclaimed their neuroprotective effects by reducing neurochemical alterations and preventing cellular dysfunction. Recent studies on AD mouse models have reported that the inhibitors of the fatty acid amide hydrolase (FAAH) and monoacylglycerol (MAGL), hydrolytic enzymes for N-arachidonoyl ethanolamine (AEA) and 2-arachidonoylglycerol (2-AG), respectively, might be promising candidates as therapeutical intervention. The FAAH and MAGL inhibitors alone or in combination seem to produce neuroprotection by reversing cognitive deficits along with Aβ-induced neuroinflammation, oxidative responses, and neuronal death, delaying AD progression. Their exact signaling mechanisms need to be elucidated for understanding the brain intrinsic repair mechanism. The aim of this review was to shed light on physiology and pathophysiology of AD and to summarize the experimental data on neuroprotective roles of FAAH and MAGL inhibitors. In this review, we have also included CB1R and CB2R modulators with their diverse roles to modulate ECS mediated responses such as anti-nociceptive, anxiolytic, and anti-inflammatory actions in AD. Future research would provide the directions in understanding the molecular mechanisms and development of new therapeutic interventions for the treatment of AD.

## 1. Introduction

In neurodegenerative diseases (NDDs), progressive loss of neuronal structure and function leads to cognitive disabilities such as dementia, neuronal death, and disruption in the homeostasis of glia [1]. NDD may arise due to age, Alzheimer’s disease (AD), Parkinson’s disease (PD), or due to genetic mutations which impact central nervous system (CNS) function: Huntington’s disease (HD) and early onset AD or PD. NDDs may predispose by multiple key processes such as aggregation of misfolded proteins, which leads to neurofibrillary tangles and plaques, cytotoxicity of neuronal cells, inflammation in response to toxic insults (e.g., protein aggregates), infection, traumatic injury, or autoimmunity [2].

The most considered process that is implicated in the CNS to protect the brain involves the immune system (microglia and macrophages) and astrocytes to remove an injurious stimulus [3]. Other characteristics, such as altered cell signaling, abnormal cell–cell communication, disrupted presynaptic input, disrupted intracellular signaling, and acquiring senescence/cell death, contribute to the pathogenesis of NDDs. Involvement of different systems and biochemical hallmarks exploit NDDs as multifactorial brain disorders. Medication that are in use mostly target single dysregulated pathways instead of targeting the multiple factors that seem responsible for NDDs, which also accounts for the lack of effectiveness in medication in slowing down the disease progression [4].

In the last few decades, the endocannabinoid system (ECS) has emerged as one of the potential molecular targets in the pathophysiology of NDDs, especially in AD to target inflammation, oxidative stress, and apoptosis responses [5]. ECS is composed of three components, i.e., (1) cannabinoid receptors (cannabinoid receptor 1-CB1R and cannabinoid receptor 2-CB2R), (2) lipid molecules synthesized from lipid membrane precursors, called endocannabinoids (eCBs), anandamide (AEA), and 2-arachidonoyl -glycerol (2-AG), and (3) enzymes that hydrolyses AEA and 2-AG, i.e., fatty acid amide hydrolase (FAAH) and monoacylglycerol lipase (MAGL). Several studies have focused on the relation between the endocannabinoid system and AD pathology [6]. To understand the structural and functional properties of this system, studies draw attention to elucidating the mechanisms that are involved in physiology and pathophysiology. Based on a literature review, these studies suggest that the modulation of ECS could compel this system back to normal functions and regain cell homeostasis [4].

This review is intended to explain the functions of ECS and eventually will discuss its physiology, pathophysiology, and later pharmaceutical interventions that have been performed so far. Our aim is also to summarize several studies conducted on animal models specifically focused on ECS to emphasize the pathophysiological conditions associated with the ECS during the progression of AD.

Moreover, this review is also driven to highlight the FAAH and MAGL hydrolytic enzyme inhibitors. These agents reduce amyloid beta (Aβ) protein deposition and inhibit dopaminergic neuronal death. FAAH and MAGL inhibitors target inflammation and anxiety in AD. Here, we did our best effort in summarizing the applications of FAAH and MAGL inhibitors in AD as potential therapeutic interventions.

## 2. Physiology of the Endocannabinoid System

The brain works in different ways by using different complex systems and keeps control over all biological functions. Most of the time, we divide those systems into the CNS and peripheral nervous system (PNS). However, other systems also work in parallel, one of the most pharmacologically important systems being ECS. This system has been gaining attention in the last few decades due to its key role it plays in CNS disorders. ECS controls and allows many biological events, such as learning, memory, emotional processing, sleep, eating, temperature control, pain, inflammation, and immune responses [7]. However, knowledge of the physiological function of the ECS is still in an emerging state and requires complete explanation.

### 2.1. Components of Endocannabinoid System

eCBs, cannabinoid receptors (CBRs), and the enzymes responsible for their biosynthesis and degradation constitute the ECS. Biological processes are controlled by endogenous signaling molecules called eCBs (AEA and 2-AG). Two members of the G-protein-coupled receptors family, CB1R and CB2R [8], are distributed with differential expressions in the human body. CB1R are primarily found in the brain areas concerned with motor control, cognition, emotional responses, motivated behavior, and homeostasis, where they control the release of neurotransmitters at the presynaptic neurons. On the other hand, CB2R is mainly located on peripheral organs, such as immune cells, monocytes, macrophages, basophils, lymphocytes, and dendritic cells, and are involved in immunological responses, control of inflammation, and other body defenses shown in Table 1.

### 2.2. Biosynthesis, Receptor’s Affinity, and Degradation

The eCBs are biosynthesized postsynaptically through a complex process after the arrival of an appropriate stimuli (Figure 1); they are known as postsynaptic retrograde mediators. The release of eCBs occurs immediately after the arrival of an adequate signal at the synaptic space through a mechanism that is not clearly understood yet. However, it seems that this mechanism is mediated by the action of endocannabinoid membrane transporter (EMT), which is supported by different studies [9,10].

Most advancement in understanding of ECS was made in early 1990s by accessing cannabis sativa extract’s endogenous target. After this study, two receptors were recognized and differentiated, and later designated as CB1R and CB2R. They are involved in physiological activity of ECS, though eCBs have different affinities and efficacies for these receptors. AEA has more affinity for the CB1R, with the fact that AEA is a partial agonist on CB1R and inactive on CB2R [11]. On the contrary, 2-AG has the same affinities for both CB1R and CB2R and acts as a full agonist on both. Few others non-CB receptors and ion channels are also identified that respond to high level of eCBs, but certainly to AEA. Interestingly, both AEA and 2-AG have been reported to interact with various other receptors as well [12].

eCBs are strictly interrelated to their reuptake by the presynaptic and postsynaptic cells and after their physiological activities on respective CBRs. Endocannabinoids are quickly degraded by enzymatic hydrolysis after their physiological activity on respective cannabinoid receptors [13].

The degradation of anandamide (AEA) is carried out by the integral membrane protein fatty acid amide hydrolase (FAAH), which breaks it down into free arachidonic acid and ethanolamine. 2-AG is primarily hydrolyzed by three enzymes in the α/β hydrolase superfamily—monoacylglycerol lipase (MAGL) and α/β-hydrolase domain-containing 6 (ABHD6) and 12 (ABHD12)—into arachidonic acid and glycerol [14]. MAGL is well characterized for its structure and function, while ABHD6 and ABHD12 remain less understood at the molecular level. In addition to enzymatic degradation, endocannabinoids can be degraded by oxidation through cyclooxygenase-2 and several lipoxygenase pathways [15,16].

As far as the termination of endocannabinoid signaling is concerned, a two-step process includes (1) transport into cells and (2) hydrolysis by two specific enzymatic systems. Both steps are rate limiting in terms of exerting a precise control over eCBs levels in tissues and rapid elimination of these signaling molecules on unnecessary conditions. eCBs uptakes are carried out by a transporter system, which is widely distributed throughout the brain, like other lipid carriers but facilitated by energy-independent manner [17].

### 2.3. ECS Retrograde Signaling

ECS functions as a lipid signaling system by the activation of endogenous small molecules—eCBs, AEA, and 2-AG function as ligands for CBRs. Normally, eCBs release in response to excitation, inhibition, or initiation of second messenger cascades by indirect effects on the glutamatergic, GABAergic, and dopaminergic systems [18]. Local excitation is mediated by retrograde signaling based on the interaction between the transmitter and receptor and release of AEA and 2-AG into the extracellular space, followed by binding with receptors localized to the presynaptic membrane [6]. Activation of CB1R in the presynapse blocks signaling of neurotransmitter release through the suppression of Ca^2+^ influx via inhibiting voltage gated Ca^2+^ channels or by inhibiting the cAMP/PKA pathway, which inhibits adenylyl cyclase. AEA has high affinity, functions as a partial agonist at CB1R, and has low affinity for CB2R, diffusing into the synaptic cleft. AEA activates CB1R and other non-CBRs, such as transient receptor potential cation channel subfamily V member 1 (TRPV1) as a full agonist that exhibits its significant role in synaptic transmission and pain regulation [19]. Moreover, AEA is also synthesized in the presynapse, and it is not clear whether AEA engages in anterograde signaling or not. Conversely, 2-AG acts as a full agonist on both CBRs (CB1R and CB2R) with moderate-to-low binding affinity. However, the interaction of 2-AG with non-CBRs has emerged recently and needs more research to expose their interactions. 2-AG is also identified to activate CB1R receptors in astrocytes [20].

The distinctive functions of eCBs seem to be a strong rationale for investigating them as therapeutic targets for NDD, including AD [21]. However, extensive research is needed to uncover the underlying mechanistic signaling pathways related to pathophysiology and cognitive decline concerned with AD, as few of them highlighted in Table 1. Therefore, a precise understanding of all the components of the ECS is necessary to unveil the pathophysiology of AD, for drug development, particularly to avoid neurotoxicity in all degenerated diseases. However, further investigation in this direction could improve the exact approach towards molecular mechanisms and development of applicable interventions for the treatment of AD, which remains a crucial unmet need.

## 3. Prospective Pathophysiology of the Endocannabinoid System (ECS)

Molecular and biochemical studies have greatly contributed to understanding the molecular mechanism that worked inside ECS during NDD pathophysiology [22]. Dysregulation at cellular level hinders the normal brain functions. However, the study of each component at a molecular level is another limitation in therapeutic regimens finding the effective prevention or halting of the neurodegeneration and repair processes [23]. Remarkable potential mechanisms and factors are responsible in the pathophysiology of AD, but are still beyond our understanding, which is the main reason for the complexity of the ECS and interconnected receptor-mediated networks [24,25]. The current findings are described in Figure 2 below, with molecular mechanisms across the synapses and the involvement of microglial and astrocyte’s CB2R and CB1R receptors, respectively. Our hypothesis is that targeting and involving the ECS in AD may be beneficial, although there are other factors, systems, and mechanisms that are not yet fully understood.

### 3.1. Functioning of Endocannabinoid Receptors (CBRs)

eCBs bind to CBRs to fulfil cellular demands. These receptors (CB1R and CB2R) perform a crucial role in ECS signaling pathways after activation with eCBs. They are G protein-coupled receptors (GPCRs), CB1R and CB2R, which interact with cellular functions, engaged with other receptors such as Transient receptor potential cation channel subfamily V member 1 (TRPV1), isolated from murine astrocytes. Another GPCR receptor—found in vascular endothelial cells that mediate local vasodilation effect of AEA; G protein-coupled receptor 119 (GPR119)—mediates some of the analgesic and anti-inflammatory effects of palmitoylethanolamide (PEA), transient receptors potential channels such as vanilloid TRPV1 receptor and Peroxisome proliferator-activated receptor alpha (PPARα) [26]. The composition of CBRs are integral membrane proteins and have a characteristic structure consisting of seven hydrophobic transmembrane domains [27].

These GPCRs are expressed at very high levels in the brain with higher protein expression levels than any other GPC receptors, N-methyl D-aspartate receptor (NMDA), or γ-aminobutyric acid type A (GABA-A) receptor [28]. CB1R is found in many different brain regions in different neuronal subpopulations and is involved in learning and memory processes [29]. Generally, they expressed presynaptically, although evidence of their presence at the somatodendritic has also been verified by different studies [30]. Moreover, CBRs are also found in peripheral tissues, including the gastrointestinal tract, reproductive tract, immune system, arteries, heart, lungs, sympathetic ganglia, and endocrine glands [31].

### 3.2. Controversial Reviews about the Neuroprotective Roles of CB1R and CB2R

CB1R are highly documented in the hippocampus and cortex regions, correlating their effects with eCBs in learning and memory processes, even though the role of CB1R in AD pathophysiology is still under discussion. In mice hippocampus, declining in CB1Rs with a rapid decline of cognitive function have noticed with the loss of neurons [32,33,34]. Some other studies showed that CB1R levels do not change in AD while suggesting their important roles in preserving cognitive functions as well [35,36,37,38]. Excitingly, CB1R together with CB2R was found with Aβ plaques in postmortem brain tissues from individuals with AD [33,39]. Several findings showed mixed results of the above investigation, and no clear evidence exists. Studies that endorse CB1R as a potential target for AD treatment still demand further investigation [40,41].

Moreover, other findings related to AD noted a parallel overexpression of eCBs and CB1R together [42,43]. Different studies at gene level suggest that gene expression and protein-targeted approaches have exhibited ECS components in hippocampus region, based on mouse model studies [44,45,46]. As the dorsal region of hippocampus is involved in memory-related functions, different expressions of CB1R have been reported in different neuronal regions, such as downregulation in glutamatergic neurons and overexpression in GABAergic neurons [47]. Furthermore, very few studies related to CB1R gene have been documented that show dysregulation of CB1R encoding gene CNR1-cannabinoid receptor 1 gene [48]. At gene level, no extensive research has been performed so far. However, understanding the complex regulation of CNR1 gene could suggest new therapeutic interventions where CB1R could be a promising target at cellular level. However, CB1R is known to have a complex expression pattern, being localized in several brain cell types (neuron/glial cells) and subcellular compartments that exhibit extremely differing behavioral functions through the activation of multiple and/or selective signaling pathways.

Activation of CB1R generally leads to hyperpolarization of neuronal membranes and stimulation of different intracellular signaling cascades [49]. CB1R mainly modulates the release of neurotransmitters, such as serotonin, glutamate, dopamine, noradrenaline, and γ-aminobutyric acid (GABA) [50]. Neuronal cells expressing CB1R are mostly GABAergic neurons and relate to the cholecystokinin positive and parvalbumin negative type of interneurons. Animal studies with induced ischemic brain injury have shown that CB1R agonists display a positive response towards neuroprotection by a reduction in glial cell damage [51].

CB2R were primarily found on endocrine immune cells and peripheral organs. On this basis, they were considered as peripheral CBR. Later, their overexpression has been noticed up to 100-fold in inflammatory processes, after tissue injuries, brain trauma, and peripheral tissue injuries, and through disrupted homeostasis [52]. CB2R has also been found on astrocytes, brainstem, and microglia under stress conditions, but not on resting microglia [53]. In activated microglia, CB2R induced anti-inflammatory cytokines though proinflammatory cytokines. This shows that immune response of CB2R plays a significant role as a key regulator in the immune system. During neuroprotection, similar responses of microglia and brain infiltrated immune cells reduce the neuroinflammation, oxidative stress, cellular apoptosis, and toxic neural excitability in AD [54].

Nevertheless, recent findings also suggest that CB1R helps regulate the immune system in AD models or traumatic brain injury. Moreover, high levels of eCBs (AEA, 2-AG) in the brain by inhibiting hydrolytic enzymes FAAH and MAGL are still a successful therapeutic option for controlling the immune response in AD and others brain disorders. Understanding AEA and 2-AG participation in relieving the symptoms of neurological diseases has become an important topic to establish new treatments.

### 3.3. Alteration in Endocannabinoid’s Expression and Involvement of FAAH and MAGL in AD

eCBs (AEA and 2-AG) exert their biological effects by binding them to receptors. The consecutive action of Ca^2+^-dependent and/or Ca^2+^-independent N-acyltransferase monitored by N-acyl-phosphatidylethanolamine specific phospholipase D seems to be the most appropriate biosynthetic pathway of AEA. 2-AG biosynthesis is regulated by either a signaling pathway of phosphatidylinositol-4,5-diphosphate or the metabolic pathway that involves triglycerides containing sn2-arachidonic esters. As these two eCBs are biosynthesized by different pathways, their concerning attributions are also different [55]. An increase in the AEA level is associated with improvements om decision-making ability and cognitive flexibility, while an increase in the 2-AG level is related to the destruction of cognitive flexibility and inhibitory response ability [55,56].

Some preclinical data have enlightened valuable effects of endocannabinoids in reducing certain neuropathological features relevant to AD. Research on animal models of AD demonstrated Aβ-induced hippocampal degeneration and cognitive decline result with a concomitant increase of 2-AG activity [57,58]. It has been concluded from different studies that endocannabinoids expression alteration may be enhanced in the AD brain as a protective mechanism against Aβ-induced damage [59,60,61]. Several studies support neuroprotective attributes of the ECS in response to wide range of neurodegeneration. Though none of these therapies in AD patients have exhibited any curative or lasting effects.These underlying pathologies are the target of many current therapies for AD [62]. 

Moreover, eCB-induced neuroprotection in response to excitotoxicity has widely been proved by various research studies and considered therapeutically beneficial for achieving different mechanisms, e.g., inhibiting glutamate release from presynaptic neurons, inhibition of Ca^2+^ release, and blockage of voltage-dependent N–, P/Q–, and L-type calcium channels [63,64]. Early inhibition of eCBs inactivation was found to reduce loss of memory retention, neuronal death, and Aβ-induced gliosis [65,66]. For eCBs inactivation, enzymes FAAH and MAGL hydrolysis the AEA and 2-AG, respectively. Because of different facts and findings, eCBs pathways are widely believed to be involved AD and FAAH has a major role in AD pathology and cognitive aspects [67,68].

FAAH and MAGL both belong to the serine hydrolase family, in which FAAH extensively distributed throughout the body with high concentrations in the brain and liver [14,69] degrades many fatty acid amides, including acylethanolamides such as anandamide and sleep factor oleamide. Although FAAH can inactivate 2-AG, the main enzyme responsible for the inactivation of 2-AG is MAGL. This enzyme is distributed in specific neuronal terminals in the brain. Besides the ground facts of hydrolytic actions of FAAH and MAGL on eCBs in ECS, their potential therapeutic applications in many CNS disorders, cancers, and neuroinflammatory diseases, many irretrievable/retrievable inhibitors have been used to explain their different selectivity.

The FAAH enzyme is mostly located in pyramidal cells of the cerebral cortex, hippocampus, and olfactory bulb [70]. This enzyme strictly controls the duration of AEA effects but to some extent also limits 2-AG activity. However, studies have revealed that MAGL mediates 85% of total hydrolyzation of 2-AG in the brain [71]. If we compare both these enzymes at a cellular level, FAAH is primarily a postsynaptic enzyme, whereas MAGL is a presynaptic inhibitor. Specific localization of both enzymes (FAAH and MAGL) also suggests that AEA and 2-AG signaling may subserve functional roles that are involved in spatial segregation [12].

### 3.4. Correlation between CB2R and FAAH

Based on a literature review and ongoing studies, a relationship exists between CB2R and FAAH in AD pathology [72]. Overexpression of CB2R and FAAH enzyme have been revealed in postmortem report of AD patient, enriched with Aβ neuritic plaque [69]. Increased FAAH activity is associated with inflammatory processes with an increase in the arachidonic acid (AA) level, the precursor of proinflammatory molecules. Therefore, increased hydrolysis of AEA in astrocytes results in increased release of AA, which further results in an inflammatory response. Consequently, FAAH inhibitors modulate the signaling of ECS and several biological responses associated with pain relief, anti-inflammatory response, and neuroprotection by acting on CB1R and CB2R [66,73]. FAAH-induced inflammation is a key problem in NDD and particularly in AD. In an in vitro experiment, knockdown of FAAH suppressed prostaglandin E2 production and pro-inflammatory gene expression. Similar results have been seen with FAAH inhibitors [74,75]. Functions of FAAH inhibitors such as the regulation of lipid metabolism and anti-inflammation have been reported in animal studies. In addition, combined effects of FAAH/MAGL inhibitors have remained a topic of interest. However, inhibition of either FAAH or MAGL enzymes has remained unable to induce a full spectrum of actions.

In AD, disrupted FAAH catalytic hydrolysis of eCBs significantly affects many physiological processes, such as neuronal protection, memory retention, cognition, pain modulation, and immune functions. Different studies have documented the major involvement of FAAH in neurodegeneration and impacts on cognitive functions. This might be considered as one of the prime targets for formulating the drug therapy for neurodegenerative diseases such as AD. Inhibition of FAAH can improve neuronal transmission, allow regulation of eCBs, and/or counteract neuroinflammation via CB1R and CB2R. Besides, it is thought that some of the pharmacological effects of dual FAAH/MAGL inhibitors are stronger than the complete inhibition of either FAAH or MAGL [76,77].

There are several evident studies that support the correlation in CB2R and inhibitory effects of FAAH and MAGL enzymes in AD pathogenesis. Increased FAAH activity is associated with inflammatory processes by increase in the level of AA, which is the precursor for proinflammatory molecules. Therefore, an increase in the hydrolysis of AEA in astrocyte cells results in high levels of AA and inflammation. Many studies reveal that systemic administration of these MAGL/FAAH inhibitors may have therapeutic impact of memory as shown in a rodent AD model [69,78].

## 4. Pharmacological Interventions for ECS in Alzheimer’s Disease

ECS components are well studied in favor of their ubiquitous distribution and differential expression in human body and CNS. This section of the review will focus on potential therapeutic targets for ECS components (Figure 3), as well as limitations in current treatments for AD. To date, the available treatments are only effective in the early stages of the disease as the etiology of AD has not been fully revealed [79]. Further investigation into the pathological mechanisms related to genes encoding CB1R and CB2R, eCBs, and metabolic enzymes involved in their synthesis and catabolism would be essential for the development of effective and safe drugs for AD.

Several studies have examined gene expression in brain regions affected by AD, such as the hippocampus, entorhinal cortex, frontal cortex, and temporal cortex, and have found that the GABAergic synapse pathway and retrograde endocannabinoid signaling pathways are thought to be involved in the altered physiological function of these regions, apart from the relatively less sensitive cerebellum [80,81,82,83]. Such studies have revealed that the GABAergic synaptic pathway, inflammatory pathways, and the retrograde endocannabinoid signaling pathways are considered or seemed to be involved in the alteration of normal physiological function of brain region in all AD-affected individuals, except the cerebellum, which is less sensitive to the effects of AD.

Furthermore, among these studies, one study has predicted three miRNAs as potential candidates targeting these genes: hsa-mir-17-5p, hsa-mir-106a-5p, and hsa-mir-373-3p [84]. Moreover, three transcription factors (TFs) were also identified as the potential upstream regulators of the robust differentially expressed genes (DEGs) in these brain regions; ELK-1, GATA1, and GATA2. No further research has been conducted under this scope for potential application of these miRNAs and TFs as therapeutic and diagnostic targets [85]. Another potential target site for AD treatment needs to be considered are TRPV channels/receptors that reduce cortex and hippocampus in AD mouse model and AD patients. TRPV1/TRPV2 have shown involvement with microglia for Aβ phagocytosis and anti-inflammatory effects in in vivo and in vitro studies [85,86]. Different approaches focused on mouse primary neuron and microglia cultures. Human datasets and AD mouse models have emphasized the correlation of TRPV1/TRPV2 expression and the ability of microglial Aβ phagocytosis [86]. Cannabidiol (CBD), the most well-known compound acting on CBRs and non-CBRs, has diverse effects on signaling pathway that interlinks microglial phagocytotic activity with TRPV2 activation. Furthermore, TRPV2 mediates PDK1/Akt-dependent phagocytosis and increased mRNA expression of phagocytosis related receptors, crucial for autophagy [87]. Moreover, CBD effectively reduces neuroinflammation by improving mitochondrial function and ATP production via TRPV2 activation [88]. Therefore, TRPV2 and other TRP channels/receptors, along with TREM2 and TREM1, could also be potential therapeutic targets in AD. CBD and other phytocannabinoids can be promising drug candidates in AD (see Table 2), though more analyses are needed.

Briefly, considering CB1R and CB2R as potential target sites for AD treatment was discussed earlier. We have found distinct and conflicting literature analysis in which CB1R expression either reduced in cortex and hippocampus regions or remain unaffected [89]. Few studies reveal that CB1R expression reduce in hippocampal and Para hippocampal areas with AD progression, while a defensive role of CB2R against neuroinflammation has been noticed due to their upregulation in microglial cells in animal models [6,90,91]. However, several research findings either support or deny CB1R involvement in AD and claim CB1R expression remains unaffected even disease progress. Nevertheless, confusingly, the activation of CB1R is associated with adverse effects on the CNS, and this has limited the use of candidate drugs that bind with CB1R [41,45]. The psychoactive adverse effects, generated by CB1R activation in the brain, limit the use of the orthosteric CB1R ligands as drugs [92,93,94]. However, the discovery of CB1R allosteric modulators in the last decade has provided new tools to target the CB1R [6,95].

CB1R is associated with adverse effects including anxiety, depression, and even suicidal attempts, and hence, the search for new therapeutic targets with minimal adverse effects is of interest. Modulation of neuroprotection by CB2R makes it a favorable topic of consideration as a potential therapeutic substrate in AD therapy [96,97,98,99]. In a normal brain, the expression of CB1R is more than CB2R. During AD, CB2R expression is upregulated that exhibits either a recovery phase or to oppose activated immunopathological conditions [100,101,102]. Predominantly, CB2R and CB1R work in contrasting conditions to regulate neuronal firing and neurotransmitter release. Studies have revealed that candidate drugs specifically acting on CB2R probably offer a novel therapeutic strategy for treating neuropsychiatric and neurological diseases without emergence of the adverse effects of CB1R [103,104]. CB2R activation results in the reduction of inflammation in response to neurotoxic and pro-inflammatory mediators by reactive astrocytes and microglial cells, modulate Aβ aberrant processing and stimulating microglial proliferation and migration [53,54,105]. Different compounds have been studied including CBD that have drawn attention to counteract Aβ-induced insults through reduction in oxidative stress, tau phosphorylation, and expression of inducible nitric oxide synthase [106,107,108].

The use of hydrolase inhibitors has been mentioned as a new drug strategy with a strong potential for treating CNS disorders [109,110]. The direct stimulation of a receptor by eCBs’ agonist or antagonist directly increases the content of eCBs. This makes the action of the agonist or antagonist on the receptor less robust than indirect stimulation and causes this strategy to be less prone to side effects. Therefore, inhibitors of FAAH and MAGL indirectly increase the excitability of the ECS by reducing the hydrolysis of endocannabinoids. Several FAAH and MAGL inhibitors have been assessed in preclinical studies, as shown in Figure 4. and summarized in Table 3.

More satisfactory results in mouse models exhibited that CBD and THC, when used in combination, their efficacy is synergized. Certainly, these studies propose that modulation of the endocannabinoid system could be an effective treatment strategy in AD.

## 5. Conclusions

We comprehensively reviewed physiological and pathological conditions of ECS and summarized the compounds under two hydrolysis enzymes that are considered the most effective therapeutics for treating AD by utilizing endocannabinoid system. In this review, we also highlighted different aspects regarding CB1R and CB2R controversial neuroprotective roles based on different conducted studies and this area needs more research to clearly understand their involvement in AD. We briefly talked about different channels and related receptors because literature surveys show limited studies on these aspects. Moreover, very little research studies are conducted on miRNA and gene expression studies; GABAergic synapse pathway and the retrograde endocannabinoid signaling pathways also demand more precise research.

We tried our best in this review particularly to collect FAAH/MAGL inhibitors and other possible therapeutics, such as endocannabinoids modulators, allosteric modulator, cannabinoids receptors agonists, antagonists, and mixed agonists. FAAH/MAGL inhibitor’s applications relating to learning and memory, neuropathic pain, anti-inflammatory, analgesic, improving synaptic plasticity, depression, anxiety, and other processes have been evaluated. Despite the above provided studies and the literature review, limited evidence supports the clinical implementation of the inhibitors in AD, and furthermore, comprehensive studies should be conducted to unveil the benefits of the potential therapeutics listed above.

After all exceeding summaries, we came to the point that pathophysiological investigations are much needed to understand the AD as well as neurological diseases mechanistic pathways, since MAGL is involved with peripheral inflammation according to many studies and only few studies have been devoted to them, those have therapeutic potential for the treatment of AD. Similar views have been associated with FAAH/MAGL dual inhibitors, the lack of existing data in animal behavior studies and experimentations gives rise to further questions that need to be solved. Compounds other than FAAH/MAGL inhibitors should be considered in future studies with pharmacological efficacy to alleviate the underlying AD pathophysiology. Moreover, comparative studies could lead to the intervention of new therapy regimens.

## Figures and Tables

**Figure 1 ijms-25-04050-f001:**
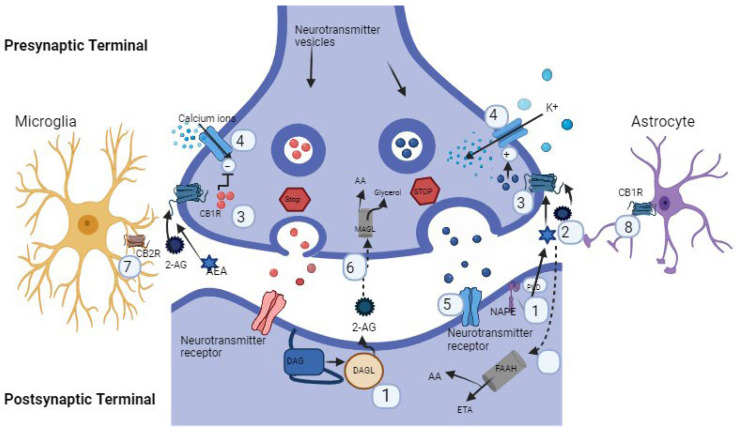
ECS retrograde signaling diagram described by using points. (1) Neurotransmitter stimulation derives ECS synthesis in postsynaptic cells. (2) ECS’s binding to CB1R on presynaptic cell. (3) Binding of ECS’s to endocannabinoid receptor results in intracellular cascade events through ion channel modulation. (4) Modulation of ion channels further ceases the release of neurotransmitters. (5) Neurotransmitter concentration in synaptic cleft affects neurotransmission in postsynaptic cell. (6) 2-AG degraded by MAGL to arachidonic acid (AA) and glycerol, while AEA metabolized in postsynaptic cell by FAAH and broken down into AA and ethanolamine (ETA). (7) Microglial CB2R interacts with 2-AG and starts immune responses. (8) Astrocytes of CB1R interact with AEA and start related effects.

**Figure 2 ijms-25-04050-f002:**
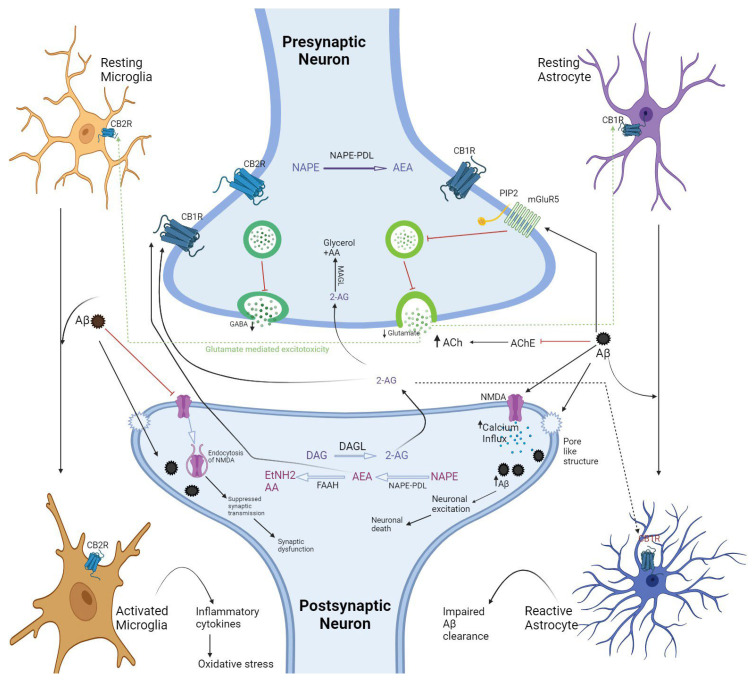
Schematic representation of prospective pathology of AD by considering ECS, summarizing many aspects regarding microglia and astrocyte, their activation on exposure to Aβ plaques. The release of AEA and 2-AG in synaptic cleft stimulates CB1R and CB2R on presynaptic neurons that further inhibit the release of neurotransmitters, such as glutamate and GABA, and cause synaptic dysfunction. Aβ is considered the most causative agent of AD because of its action in microglia and astrocyte activation, which lead to neuronal damage through glutamate-mediated excitotoxicity. Aβ enhances the influx of calcium ions through postsynaptic NMDA receptors or pore-like structure; it also inhibits AChE and increases the Ach level in the synaptic cleft, which further increases postsynaptic calcium influx. This results in downstream signaling pathways related to neuronal excitotoxicity (decrease glutamate). Ethylamine (EtNH_2_), N-acyl phosphatidylethanolamine phospholipase D (NAPE-PDL), Phosphatidylinositol bisphosphate.

**Figure 3 ijms-25-04050-f003:**
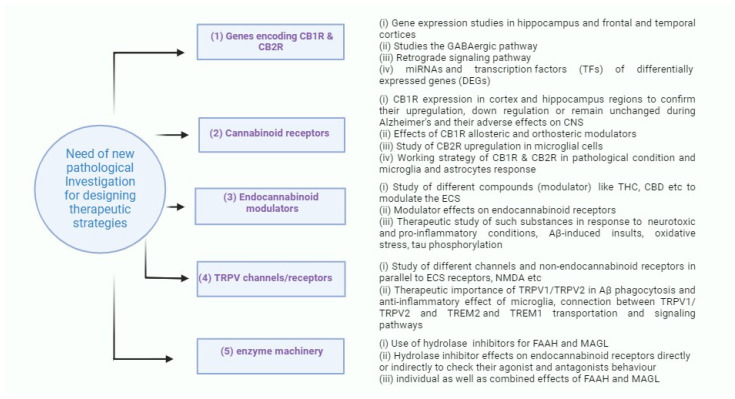
Prospective ways of investigation concerned with different pathological lines for designing potential therapeutic strategies.

**Figure 4 ijms-25-04050-f004:**
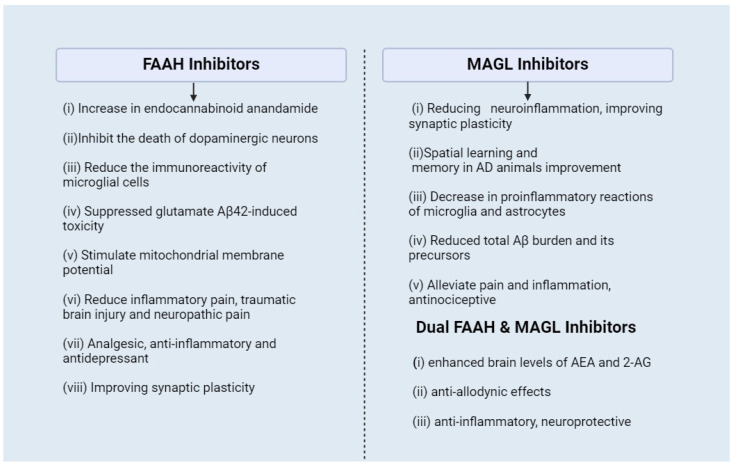
Effects of FAAH inhibitors, MAGL inhibitors, and dual FAAH and MAGL inhibitors at neuronal terminals and their different pharmacological effects.

**Table 1 ijms-25-04050-t001:** Summarize the connection between eCBs and AD and effects on up/downregulation on ECS.

Aspects	Summary
eCBs and alzheimer’s	no conclusive evidence that cannabis or eCBs can stop, reverse, or prevent dementia.
Behavioral symptoms	some studies suggest eCBs may help manage agitation and aggression in dementia patients. However, long-term effects are unclear.
Lab studies	endocannabinoids appear to remove amyloid protein (Alzheimer’s hallmark) from nerve cells in some lab studies.
Clinical trials	no specific clinical trials on endocannabinoids in dementia patients.
ECS and CB1R	ECS plays a role in neurotransmission and neuroimmune systems. CB1R are abundant in the brain and hypothalamus, modulating endocrine axes.
ECS and CB2R	CB2R are mainly expressed in immune cells.
effects of CB1R agonists	may help manage symptoms such as agitation and anxiety in dementia.
Effects of eCBs antagonists	antagonists can inhibit CB1R activation by endogenously released eCBs. They may also act as inverse agonists, shifting CB1R activity from an active state to an inactive state.
Upregulated or downregulated	moderate activation of CB1R by selective agonists or eCBs may have acute beneficial therapeutic actions, such as pain relief and anti-psychotic effects. Elevation of eCBs levels by inhibiting degrading enzymes (such as FAAH and MAGL) can also be beneficial.
Neuroprotective properties	ECs may have neuroprotective properties, potentially reducing the risk of neurodegenerative diseases associated with dementia.

**Table 2 ijms-25-04050-t002:** Overview of different compounds (with therapeutic effects) acting as endocannabinoids modulators, allosteric modulator, cannabinoids receptors agonists, antagonists, and mixed agonists.

AEA and 2-AG Modulators	CB1R Specific Agonists	CB1R Specific Antagonists	Allosteric Modulators of CB1R	CB2R Specific Agonists	CB2R Specific Antagonists	Mixed Receptors Agonist
URB597 (symptoms of schizophrenia)	ACEA (anti-inflammatory)	Rimonabant (SR141716) (obesity, type 2 diabetes, dyslipidemia)	Org27569-positive allosteric modulator (PAM)	JWH-133 (reduction in inflammatory components)	SR144528 (anti-nociceptive)	THC (inflammatory pain, metabolic syndrome, obesity))
PF-04457845 (osteoarthritic pain)	ACPA (anti-inflammatory)	AM-251 (decrease excitotoxicity)	Pepcan-12-negative allosteric modulator (NAMs)	JWH-051 (relative lack of psychotropic effects compared to CB1R agonist)	AM-630 (pain)	CBD (neuroinflammatory diseases, chronic and inflammatory pain)
JZL184 (anti-nociceptive)	Methanandamide (neuroprotection)	AM-281 (neuroprotection)		PM-226		WIN-55, 212–2
JZL195 (anti-nociceptive)	O-1812 (Neuroprotection)	Taranabant (MK-0364)		HU-308		Nabilone (prevent memory loss)
		JD5037-peripheral CB1R inverse agonist (neither affects behavioral responses mediated by CB1Rs in the brain)		GW-405833 (pain)		JWH-018
		AM4113		L-759, 633		HU-210 (anti-inflammatory, anti-psychotic)
		LH-21		L-759, 656		CP-55, 940 (anti-emetic)
		PIMSR		O-1966 AM1241 Xie2–64 AM1710 JHW015 (specific CB2R agonists in CNS disorder)		
				MDA7		

**Table 3 ijms-25-04050-t003:** Therapeutic effects of FAAH and MAGL compounds inhibiting endocannabinoid degradation based on preclinical studies (IC50: 50% inhibitory concentration; h: human; r: rat; m: mouse; data are provided as mean, mean ± standard deviation).

Class	Compounds	IC50 (nM)	Therapeutic Effects	References
FAAH inhibitor (irreversible covalent)	URB597	33.5 (h) 3.8 (r) 45 (m)	(i) increase in endocannabinoid anandamide, (ii) suppressed glutamate Aβ42-induced toxicity (iii) stimulate mitochondrial membrane potential (iv) reduction in interleukin (IL)-1β (v) tumor necrosis factor-α (TNFα) expression (vi) neuropathic pain	[111,112]
	URB937	26.8 ± 4.9 (r)	(i) increase in endocannabinoid anandamide, (ii) inflammatory pain (iii) neuropathic pain	[113]
	AM374	13 (r)	(i) increase in endocannabinoid anandamide	[114]
	AM3506	2.8 ± 0.3 (r)	(i) hypertension (ii) posttraumatic stress disorder	[115]
	PF-3845	18 (h)	(i) inflammatory pain (ii) traumatic brain injury (iii) neuropathic pain (iv) visceral pain (v) anxiety-related disorders	[116]
	PF-04457845 (PF-7845)	7.2–50.4 (h) 7.4–43.1 (r) 2.5 (m)	(i) analgesic and anxiolytic effects (ii) 25-fold higher than URB597 for high in vivo efficacy (iii) long duration of action for inflammatory pain (iv) very well tolerated in healthy subjects	[117]
	PF-750	16.2-595	(i) analgesic, (ii) anti-inflammatory, (iii) anti-depressant	[114,118]
	LY2183240	37.3 ± 5.4 (h) 2.1 (r) 12.4 (m)	(i) analgesic	[118,119,120]
	MAFP	0.33 ± 0.07 (h) 2.5 (m) 1–3 (r, m)		[121,122,123]
	SA-57	1.9 (h) 3.2 (m)		[124,125]
FAAH inhibitor (reversible binding)	OL-135	206 (h) 2.1 (m)	(i) neuropathic pain (ii) inflammatory pain (iii) visceral pain	[114,122,126]
	OL-92	0.28 (m)		[127]
	MK-4409	11 (h) 11 (r)	(i) neuropathic pain	[110,128,129]
	ST4070	9 (m)		[130]
FAAH inhibitors (slowly reversible binding)	JZP-327A	11 (h)		[73,131]
FAAH inhibitor (partial reversible binding)	JNJ-1661010	33 ± 2.1 (h) 34 ± 6.5 (r)	(i) neuropathic pain (ii) inflammatory pain	[132,133]
FAAH inhibitor (partial irreversible inhibitor)	JNJ-42165279	313 ± 28 (r)	(i) analgesic and anxiolytic effects	[114,134]
FAAH inhibitor (reversibility not available)	URB880	0.63 ± 0.04 (r)	(i) analgesic (ii) anti-depressant	[135]
irreversible binding with MAGL	N-arachidonoyl maleimide	2180 ± 620 (h) >10,000 (r)	(i) reducing neuroinflammation (ii) improving synaptic plasticity (iii) spatial learning (iv) memory in AD animals	[136,137]
	Disulfiram	360 (h) (surrogate substrate assay)	(i) reducing neuroinflammation (ii) improving synaptic plasticity (iii) spatial learning (iv) memory in AD animals	[138]
	SAR-629	0.9 (h) 0.22 (m)	(i) reducing neuroinflammation (ii) improving synaptic plasticity (iii) spatial learning (iv) memory in AD animals	[139]
	JJKK-006	0.6 (h)	(i) reducing neuroinflammation (ii) improving synaptic plasticity (iii) spatial learning (iv) memory in AD animals	[140]
	JJKK-048	0.363 (h) 0.214 (r) 0.275 (m)	(i) reducing neuroinflammation (ii) improving synaptic plasticity (iii) spatial learning (iv) memory in AD animals	[140,141]
	ML-30	0.54 (h) 4.4 (r) 1.9 (m)	(i) reducing neuroinflammation (ii) improving synaptic plasticity (iii) spatial learning (iv) memory in AD animals	[139,140]
	KML-29	5.9 (h) 43 (r) 15 (m)	(i) reducing neuroinflammation (ii) improving synaptic plasticity (iii) spatial learning (iv) memory in AD animals	[141]
	JZL-184	3.9 (h) 262 (r) 10 (m)	(i) decrease in proinflammatory reactions of microglia and astrocytes. (ii) reduced total Aβ burden and its precursors	[142,143]
	JW642	4.7 (h) 14 (r) 7.6 (m)	(i) anti-hyperalgesia (ii) anti-anxiety/depression	[142]
reversible inhibitor of MAGL	JZP-361	46 (h)	(i) alleviate pain and inflammation	[142]
slowly reversible with MAGL	AM6701	1.2 ± 0.35 (h) 1.7 (r)	(i) neurodegenerative cascade, Behavioral deficits linked to Seizure’s damage	[144,145]
partially reversible binding with MAGL	URB-602	360 (h) 28,000 (r)	(i) anti-nociceptive, anxiolytic, anti-inflammatory	[146]
MAGL/FAAH dual inhibitor	JZL-195	4 (hMAGL) 2 (hFAAH) 19 (mMAGL) 13 (mFAAH)	(i) enhanced brain levels of anandamide and 2-AG (ii) anti-allodynic effects	[139,142,147]
irreversible inhibitor of both MAGL and FAAH enzymes	CAY10499	76 (h) 86 (r)	(i) anti-inflammatory (ii) neuroprotective	[147]

Note: FAAH and MAGL inhibitors are differentiated with one another based on their IC50 values which show the potency of inhibition. The above lists under respective categories show their therapeutic effects that have been documented in different studies related to AD.

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
