# Peer review of "Potential Therapeutic Targets to Modulate the Endocannabinoid System in Alzheimer’s Disease"

_ijms, 2024, doi:10.3390/ijms25074050_

Round 1

Reviewer 1 Report

Comments and Suggestions for Authors

1. Figure 2 should be described in detail with reference to bibliographic sources

2. In my opinion the Authors should summarise in a table any information that indicates a link between the endocannabinoid system and the AD, as such information is presented in many subsections, but it is useful to extract the most important possible connections. In this aspect, I suggest to indicate what part of ECS I up/down-regulated, whether agonist or CBR antagonists induce any positive effect, etc.

3. Please elaborate on all the abbreviations of the compounds in the figure 2.

4. Since the Authors introduce a table showing various kinds of FAAH and or MAGL inhibitors, and divide them into reversible or nonreversible, partial or full inhibitors, I suggest to provide some additional information regarding some of examples of the compounds. In line with this, it would be worth to mention whether partial or full, reversible or non may have a better efficacy in AD treatment. Also, what type od adverse effects can be seen after administration of such compounds. Are there any clinical data?

5. Please provide the limitation of the study presented

Comments on the Quality of English Language

moderate changes are required

Author Response

Hi,

Many thanks and I highly appreciate your suggestions and comments related to figure and tables.

1) Your concerns about figure. 2 was nice and I used to elaborate the figure more and abbreviations as well.

2) As far as your comment 2 is concerned I have already explain it under Highlight 3. "Prospective Pathophysiology of ECS". Moreover, I have added a bit more.

3) Related to your comment 4, the list of FAAH and MAGL are categorized with their different range of activities. If there any clinical data was available, I mentioned it with reference.

Well, I tried my best to make changes and modifications according to your suggestions.

Thanks again.

Hina Kanwal

Reviewer 2 Report

Comments and Suggestions for Authors

The manuscript is comprehensive and well-organized. I have a few suggestions to make, please provide a search strategy used for data collection and screening for the review. It was good to mention the type of review ' narrative review' in the introduction.

2. Figure 3 quality can be improved as it is blurred.

3. Clinical application of the results can be discussed before the conlcusion.

Comments on the Quality of English Language

Minor revision and editing are required before publishing

Author Response

Hi,

Many thanks for your time to review our review article and your kind suggestions that I really appreciate it.

I have replaced the figure. 3 with a better resolution one.

As far as, you are asking for clinical application, there are very few studies on it so I have added some limitation text in that reference.

Thanks again.

Best,

Hina Kanwal

Round 2

Reviewer 1 Report

Comments and Suggestions for Authors

Dear Authors, thank you for your response to the review. Obviously, the paper has been improved. However, still I insist to provide a table summarizing the most important info on the link between CBs and AD. I see that such data are widely presented in the section 3, though it would be worth to prepare and show a table. So, again I suggest the Authors to extract the most important possible connections. In this aspect, I suggest to indicate what part of ECS I up/down-regulated, whether agonist or CBR antagonists induce any positive effect,

Author Response

Hi Editor,

Thanks again for pointing out, it will definitely help our review to be more concise to summarize the materials in a table form. 

I really appreciate it. I have provided a summary on the connection between endocannabinoids and AD and related effects on up/downregulation on endocannabinoid system.

Thanks for making us more precise.

Best,

Hina Kanwal